# Pyrethroid and Etofenprox Resistance in *Anopheles gambiae* and *Anopheles coluzzii* from Vegetable Farms in Yaoundé, Cameroon: Dynamics, Intensity and Molecular Basis

**DOI:** 10.3390/molecules26185543

**Published:** 2021-09-13

**Authors:** Michael Piameu, Philippe Nwane, Wilson Toussile, Konstantinos Mavridis, Nadja Christina Wipf, Paraudie France Kouadio, Lili Ranaise Mbakop, Stanislas Mandeng, Wolfgang Eyisap Ekoko, Jean Claude Toto, Kelly Lionelle Ngaffo, Petronile Klorane Ngo Etounde, Arthur Titcho Ngantchou, Mouhamadou Chouaibou, Pie Müller, Parfait Awono-Ambene, John Vontas, Josiane Etang

**Affiliations:** 1Laboratoire de Recherche sur le Paludisme, Institut de Recherche de Yaoundé (IRY), Organisation de Coordination pour la lutte Contre les Endémies en Afrique Centrale (OCEAC), P.O. Box 288, Yaoundé 999108, Cameroon; piameujr@yahoo.fr (M.P.); philino07@yahoo.fr (P.N.); wilson.toussile@gmail.com (W.T.); mbalira@yahoo.fr (L.R.M.); mandengelysee@yahoo.fr (S.M.); eekokowolfgang@yahoo.com (W.E.E.); jctotofr@yahoo.fr (J.C.T.); hpaawono@yahoo.fr (P.A.-A.); 2Ecole des Sciences de la Santé, Université Catholique d’Afrique Centrale, P.O. Box 1110, Yaoundé 999108, Cameroon; p.klorane@yahoo.com (P.K.N.E.); arthurngantchou12@gmail.com (A.T.N.); 3Department de Biologie et Physiologie Animales, Faculté des Sciences, Université de Yaoundé I, P.O. Box 812, Yaoundé 999108, Cameroon; 4Centre de Recherche sur les Filarioses et Maladies Tropicales (CRFilMT), P.O. Box 5797, Yaoundé 999108, Cameroon; 5Département de Mathématiques et Sciences Physiques (MPS), Ecole Nationale Supérieure Polytechnique de Yaoundé (ENSPY), Université de Yaoundé 1, P.O. Box 8390, Yaoundé 999108, Cameroon; 6Institute of Molecular Biology and Biotechnology, Foundation for Research and Technology-Hellas, 70013 Heraklion, Greece; mavridiskos@gmail.com (K.M.); vontas@imbb.forth.gr (J.V.); 7Swiss Tropical and Public Health Institute, Socinstrasse 57, 4002 Basel, Switzerland; nadja.wipf@swisstph.ch (N.C.W.); pie.mueller@swisstph.ch (P.M.); 8University of Basel, Petersplatz 1, 4001 Basel, Switzerland; 9Centre Suisse de Recherches Scientifiques en Côte d’Ivoire, P.O. Box 1303, Abidjan 1303, Cote d’Ivoire; paraudiek@gmail.com (P.F.K.); mouhamadou.chouaibou@csrs.ci (M.C.); 10Laboratory of Animal Biology and Physiology, Faculty of Sciences, University of Yaoundé I, P.O. Box 337, Yaoundé 999108, Cameroon; 11Laboratory of Animal Biology and Physiology, University of Douala, P.O. Box 24157, Douala 999108, Cameroon; 12Institut de Recherche en Sciences de la Santé (IRSS), Centre d’excellence Africain en Innovations Biotechnologiques pour l’élimination des Maladies à Transmission Vectorielle (CEA/ITECH-MTV), Université Nazi Boni, P.O. Box 545, Bobo-Dioulasso 22620, Burkina Faso; kellylionelle@yahoo.fr; 13Department of Crop Science, Agricultural University of Athens, Iera Odos 875, 11855 Athens, Greece; 14Department of Biological Sciences, Faculty of Medicine and Pharmaceutical Sciences, University of Douala, P.O. Box 2701, Douala 999108, Cameroon; 15Institute for Insect Biotechnology, Justus-Liebig-University Gießen, 35394 Gießen, Germany

**Keywords:** malaria, insecticides, vector control, urban areas, resistance intensity, *Anopheles gambiae*, *Anopheles coluzzii*, Cameroon

## Abstract

Previous studies have indicated widespread insecticide resistance in malaria vector populations from Cameroon. However, the intensity of this resistance and underlying mechanisms are poorly known. Therefore, we conducted three cross-sectional resistance surveys between April 2018 and October 2019, using the revised World Health Organization protocol, which includes resistance incidences and intensity assessments. Field-collected *Anopheles gambiae* s.l. populations from Nkolondom, Nkolbisson and Ekié vegetable farms in the city of Yaoundé were tested with deltamethrin, permethrin, alpha-cypermethrin and etofenprox, using 1× insecticide diagnostic concentrations for resistance incidence, then 5× and 10× concentrations for resistance intensity. Subsamples were analyzed for species identification and the detection of resistance-associated molecular markers using TaqMan® qPCR assays. In Nkolbisson, both *An. coluzzii* (96%) and *An. gambiae* s.s. (4%) were found together, whereas only *An. gambiae* s.s. was present in Nkolondom, and only *An. coluzzii* was present in Ekié. All three populations were resistant to the four insecticides (<75% mortality rates―MR1×), with intensity generally fluctuating over the time between mod-erate (<98%―MR5×; ≥98%―MR10×) and high (76–97%―MR10×). The *kdr* L995F, L995S, and N1570Y, and the *Ace-1* G280S-resistant alleles were found in *An. gambiae* from Nkolondom, at 73%, 1%, 16% and 13% frequencies, respectively, whereas only the *kdr* L995F was found in *An. gambiae* s.s. from Nkolbisson at a 50% frequency. In *An. coluzzii* from Nkolbisson and Ekié, we detected only the *kdr* L995F allele at 65% and 60% frequencies, respectively. Furthermore, expression levels of *Cyp6m2*, *Cyp9k1*, and *Gste2* metabolic genes were highly upregulated (over fivefold) in Nkolondom and Nkolbisson. Pyrethroid and etofenprox-based vector control interventions may be jeopardized in the prospected areas, due to high resistance intensity, with multiple mechanisms in *An. gambiae* s.s. and *An. coluzzii*.

## 1. Background

The intensification of core vector control interventions with long-lasting insecticidal nets (LLINs) and indoor residual spraying (IRS), associated with the improved diagnosis and treatment of cases has resulted in substantial reductions in the global malaria incidence between 2000 and 2015 [1,2]. Indeed, an estimated 663 million clinical cases were averted during this fifteen-year period, with 68%, 22% and 10% contributions by LLINs, artemisinin combination therapies (ACTs) and IRS, respectively [1]. However, between 2015 and 2019, the decline in malaria incidence stalled to less than 2%, compared to 27% between 2000 and 2015 [2]. In 2019, the number of cases was globally estimated at 229 million in 87 endemic countries. At the same time, the rapid expansion of insecticide resistance in the most efficient *Anopheles* vectors of *Plasmodium* parasites jeopardizes the interventions’ effectiveness. Indeed, insecticide resistance of *Anopheles* populations to at least one insecticide class has been reported in 73 malaria-endemic countries [2]. In 28 countries, malaria vector populations are resistant to the four main insecticide classes used in public health control (i.e., carbamates, organochlorides, organophosphates and pyrethroids) [2]; of particular concern is the resistance to pyrethroid insecticides, because all World Health Organization (WHO) prequalified LLINs contain at least one pyrethroid [3].

Mosquito populations displaying resistance to public health insecticides are very common in areas with agricultural cultivations [4,5,6,7] and in locations with intensive insecticide-based vector control interventions [8,9], presumably due to the high selective pressure imposed by the insecticides deployed. In addition, the accumulation of waste and other pollutants generated through human activities in the environment around mosquito breeding sites tends to increase the selection of mosquito resistance to insecticides [10].

In Cameroon, malaria prevention largely relies on the use of LLINs [11]. During the last decade, over 20 million LLINs were freely distributed to the general population through several nationwide campaigns, which resulted in 77% of the population owning at least one treated net, and 58% of the population using LLINs regularly [11,12]. As a result of these efforts, the prevalence of malaria in the general population decreased from 41% to 24% between 2000 and 2015, and the related mortality decreased by 54% (i.e., from about 13,000 to 6000) [11,12]. Alternative measures, such as IRS and larviciding, are still in the pilot phase [12].

Despite the LLIN campaigns, Cameroon remained one of the 11 countries accounting for 70% of the global burden of malaria in 2019 [2]. The factors leading to high malaria burden despite the interventions are poorly known. Meanwhile, the country is committed to a high-burden/high-impact approach. Regarding the insecticide susceptibility of the malaria vectors, studies conducted during the last 20 years on the major malaria vector species in Cameroon, including *An. gambiae* s.s., *An. coluzzii* and *An. arabiensis*, have shown that pyrethroid resistance has rapidly increased since its first report in 2003 [13,14,15]. The primary mechanisms conferring pyrethroid resistance in *An. gambiae* s.s. and *An. coluzzii* are target-site point mutations in the para voltage-gated sodium channel gene, including the knockdown resistance (*kdr*) L995F/S (formerly known as L1014F/S) and N1570Y (formerly known as N1575Y) mutations [14,15,16,17]. In addition, the increased expression of cytochrome P450 monooxygenases (P450) *CYP6P3*, *CYP6M2*, and *CYP6P4*, and glutathione *S*-transferase (GST) *GST1-6* has been associated with pyrethroid resistance [18,19]. Furthermore, the acetylcholinesterase mutation *Ace-1* G280S (formerly known as *Ace-1* G119S) that confers resistance to carbamates and organophosphates has been detected in the same mosquito species in Cameroon [18,19,20].

A previous study attempting to assess the intensity of deltamethrin resistance in *An. gambiae* s.l. from Pitoa in North Cameroon using dose-response assays revealed a more than 500-fold resistance ratio compared to the *An. gambiae* s.s. Kisumu insecticide-susceptible reference strain [21]. Furthermore, exposing these mosquitoes to a 10× deltamethrin diagnostic concentration (0.5%) displayed a moderate level of resistance according to WHO classifications [21]. Similarly, a longitudinal study based on the WHO insecticide susceptibility assay conducted over 5 years from 2011 to 2015 confirmed the rapid expansion of deltamethrin resistance in *An. gambiae* s.l. populations from the North region of Cameroon, especially in urban settings and agricultural areas [17]. All these findings highlight the variability and complexity shaping insecticide resistance patterns in *An. gambiae* s.s., *An. coluzzii* and *An. arabiensis* populations from Cameroon. However, data on the intensity of insecticide resistance in various ecological settings in Cameroon and its potential operational significance are very limited. Therefore, in-depth investigations of insecticide resistance are essential for identification of the areas where alternative or complementary vector control interventions are urgently needed on top of LLINs, e.g., scaling up IRS or larviciding.

The aim of the current study was to assess the dynamics of pyrethroid resistance in terms of incidence and intensity in *An. gambiae* s.l. populations from three vegetable farming areas in the city of Yaoundé, namely, Nkolondom, Nkolbisson and Ekié (Figure 1). The second objective was to assess the susceptibility of pyrethroid resistant *An. gambiae* s.l. populations to etofenprox, as a potential active ingredient for LLINs and IRS in pyrethroid resistance areas. We sampled mosquito larvae from the field in three cross-sectional surveys between April 2018 and October 2019, reared them until adult stage in the laboratory, assessed the insecticide resistance incidence and intensity using WHO insecticide susceptibility bioassays, and then we used molecular assays to identify the underlying resistance mechanisms. The study was conducted during the short rainy season (April–May 2018), the main dry season (December 2018–January 2019) and the main rainy season (September–October 2019).

## 2. Results

### 2.1. Species Composition

In April–May 2018, 50 adult mosquito specimens from each study site emerging from field-collected larvae and morphologically identified as belonging to the *An. gambiae* complex were further identified to species level, using TaqMan^®^ qPCR assays. In addition, 50 specimens from an *An. gambiae* s.s. insecticide-susceptible laboratory colony (the Kisumu strain) were included as a control in the analysis. Overall, two sibling species, *An. gambiae* s.s. and *An. coluzzii*, were identified. Both species were found in Nkolbisson, with a predominance of *An. coluzzii* (96%), whereas in Nkolondom and Ekié, all specimens were *An. gambiae* s.s. and *An. coluzzii*, respectively (Table 1). Mosquito samples of the Kisumu strain were confirmed as *An. gambiae* s.s.

### 2.2. Trends of Insecticide Resistance in Anopheles gambiae s.l. Populations

A total of 9582 *An. gambiae* s.l. specimens from Ekié (*n* = 3156), Nkolbisson (*n* = 3215) and Nkolondom (*n* = 3211) were used for insecticide susceptibility testing. The bioassays also included 456 adult females of the Kisumu *An. gambiae* s.s. reference susceptible colony as a negative control. Overall, 111 bioassays were performed during three consecutive surveys in April–May 2018, December 2018–January 2019 and September–October 2019. These included 99 susceptibility tests with field-collected *An. gambiae* s.l. specimens and 12 tests with the Kisumu reference colony. For each study site, 32–34 tests were carried out (Figure 2). These included:

Twelve susceptibility tests using insecticide diagnostic concentrations (1× DCs) (i.e., 0.05% deltamethrin, 0.05% alpha-cypermethrin, 0.75% permethrin and 0.5% etofenprox);

Twelve insecticide resistance intensity tests using four 5× DC (i.e., 0.25% deltamethrin, 0.25% alpha-cypermethrin, 3.75% permethrin and 2.5% etofenprox);

Finally, 8–10 insecticide resistance intensity tests using 10× DC (i.e., 0.5% deltamethrin, 0.5% alpha-cypermethrin, 7.5% permethrin and 5% etofenprox), depending on the availability of mosquitoes.

#### 2.2.1. Resistance Frequencies

Mortality rates in the controls, exposed to silicone-oil-impregnated paper only, were below 5%, whereas the mortality rates for the Kisumu colony against all insecticides were consistently above 98% (Table 2). In contrast, across the three study sites, the field populations showed resistance to all four insecticides tested (Figure 3). The lowest mortality rates were recorded with permethrin (mostly less than 20%), whereas the three other insecticides induced variable mortality rates ranging from 25% to 75%, suggesting temporal or season fluctuations of insecticide resistance.

Depending on the study site, different patterns of time-based variations of mortality rates were observed. In samples from Nkolondom, the mortality rates to deltamethrin and etofenprox gradually increased, from less than 20% in April–May 2018 to around 40% in September–October 2019, suggesting a temporal decrease in resistance frequency to these insecticides. In contrast, the mortality to permethrin decreased from 20% to less than 5% in the same period, suggesting an increase in resistance frequency to this insecticide. However, the mortality to alpha-cypermethrin was higher during the dry season in December 2018 (70%) compared with the two rainy seasons (around 10%), suggesting very important fluctuations of alpha-cypermethrin resistance frequency from one season to another.

In *An. gambiae* s.l. samples from Nkolbisson, the mortality rates to deltamethrin gradually decreased, from 70% in April–May 2018 to less than 15% in September–October 2019, suggesting a progressive increase in resistance frequencies over the time. However, the mortality rates to permethrin and alpha-cypermethrin fluctuated between 15% and 30% in April–May 2018 and December 2018–January 2019; then, they decreased to less than 10% in September–October 2019, suggesting a deferred increase in resistance frequencies to these insecticides. In contrast, the mortality to etofenprox increased from 20% to 75% over time, suggesting a decrease in etofenprox resistance frequency.

In *An. gambiae* s.l. samples from Ekié, seasonal variations of resistance frequencies were observed for deltamethrin and etofenprox. Indeed, the lowest mortality to deltamethrin was recorded during the dry season (≈30%) and the highest (≈70%) during the rainy seasons. An opposite propensity was seen with etofenprox, against which mosquito mortality was significantly higher during the dry season (≈30%) compared with the rainy seasons (<20%) (*p* < 0.05%). For permethrin and alpha-cypermethrin, mosquito mortality rates were mostly less than 20%, although a significant increase was seen with alpha-cypermethrin in October 2019 (40%) (*p* < 0.05%).

#### 2.2.2. Status and Dynamics of Resistance Intensity

The mortality rates of *An. gambiae* s.l. from Nkolondom, Nkolbisson and Ekié resulting from resistance intensity tests are provided in Figure 4. Against the 5× DCs, fewer than 95% of the mosquitoes were killed, irrespective of the survey period or the study site, indicating that the intensity of resistance in surveyed mosquito populations was either moderate or high. Indeed, the use of 10× DC confirmed moderate or high resistance in *An. gambiae* s.l. from the three study sites, depending on the survey period and the insecticide.

Overall, the resistance intensity was moderate (<98% MR_5×_; ≥98% MR_10_) in 11 of the 36 assays and high (45–97% MR_10×_) in 25 of the assays. Considering the wide range of mortality rates indicating high resistance intensity (according to the WHO criteria), we further classified high resistance intensity into three levels: level 1 (75–97% mortality rate―HL_1_), level 2 (50–74% mortality rate―HL_2_) and level 3 (<50% mortality rate―HL_3_) (Table 2). According to this further classification, the high resistance intensity was mostly at HL_1_ (in 21/25 tested samples, 84%), with only four cases at HL_2_ and HL_3_ (4/25 tested samples, 16%). Both HL_2_ and HL_3_ resistance intensity to deltamethrin were recorded in mosquito samples from Nkolbisson, as well as those from Nkolondom to alpha-cypermethrin.

Depending on study sites and insecticides used, changes in resistance intensity were observed. The intensity of deltamethrin resistance stalled at HL_1_ in Nkolondom and at moderate level (MD) in Ekié, whereas it progressively increased from HL_1_ in April–May 2018 to HL_2_ (50–75% mortality rates) in December 2018–January 2019 and HL_3_ (<50% mortality rates) in September–October 2019 for mosquito samples from Nkolbisson.

With permethrin, the intensity of resistance changed in the three mosquito populations during the December 2018–January 2019 study period, either from MD to HL_1_ or vice versa. With alpha-cypermethrin, the intensity of resistance stalled at HL_1_ in mosquito samples from Nkolbisson, whereas it progressively increased from HL_1_ in April–May 2018, to HL_2_ (50–75% mortality rates) in December 2018–January 2019 and HL_3_ (<50% mortality rates) in September–October 2019 for mosquito samples from Nkolondom.

However, the intensity of resistance to alpha-cypermethrin in mosquito samples from Ekié decreased from HL_1_ in December 2018–January 2019 to MD in September–October 2019. Additionally, at the same period, the intensity resistance to etofenprox was minimized in the three study mosquito populations, from HL_1_ to MD.

#### 2.2.3. Relationship between the Frequency and the Intensity of Insecticide Resistance

A broad analysis of the mortality rates of *An. gambiae* s.l. samples post-exposure to 1× DC and 10× DC (Table 2) revealed a wide variation in mortality rates to the 1× DC, falling under the threshold of confirmed resistance. However, the corresponding resistance intensity to 10× DC was mostly either MD or HL_1_, except for a few cases of HL_2_ and HL_3_ recorded in Nkolbisson and Nkolondom in September–October 2019 (<40% mortality rates to 10× deltamethrin and alpha-cypermethrin DC, respectively). The HL_3_ resistance intensity to 10× DC was associated with mortality rates of less than 5% to 1× DC. However, for the MD, HL_1_ and HL_2_ resistance intensities, the counter mortality rates post-exposure to the 1× DC varied widely, from 0.0% to 75%.

### 2.3. Molecular Markers of Insecticide Resistance

Overall, there was a site-to-site variation in the allelic frequencies of resistance molecular markers (Table 1 and Table 3).

In the *An. gambiae* s.s. population from Nkolondom, the *kdr* L995F allele was recorded in the highest frequency (72.0%). Interestingly, the *kdr* L995S allele was also found, although at a heterozygote state with L995F in only one mosquito specimen (1.00% allelic frequency). Nkolondom was also the only area where the *kdr* N1570Y allele was recorded (at a frequency of 16.0%). The *Ace-1* G280S allele was also found at a 13.0% frequency.

In Nkolbisson, the *kdr* L995F allele was present at a frequency of 50.0% in *An. gambiae* s.s. and 65.0% in *An.*
*coluzzii*. Neither mutant *kdr* L995S, N1570Y nor *Ace-1* G280S alleles were recorded in either *An. gambiae* or *An.*
*coluzzii*.

In the *An. coluzzii* population from Ekié, the *kdr* L995F allele was found at a 60.0% frequency, whereas no mutant *kdr* L995S, N1570Y or *Ace-1* G280S were detected.

The gene expression analysis of genes putatively implicated in insecticide resistance measured in the Nkolondom and Nkolbisson *An. gambiae* s.l. populations showed that the following genes were upregulated compared to the susceptible Kisumu laboratory colony: *Cyp6m2*, *Cyp9k1*, *Cyp6p4*, *Cyp6z1*, G*ste2* and *Cyp4g16* (Table 3). The genes with the highest fold changes (>5-fold) in the Nkolondom population were the P450s *Cyp6m2* (17.1-fold) and *Cyp9k1* (6.0- fold) and the GST *Gste2* (12.8-fold). *Cyp6m2* and *Cyp9k1* were also among the most upregulated genes in the Nkolbisson population together with *Cyp6p4* (6.2-fold) and *Gste2* (52.0-fold).

## 3. Discussion

In this study, a stepwise assessment of insecticide resistance in *An. gambiae* s.l. populations from three urban areas in Yaoundé provided evidence for moderate- to high-intensity resistance to pyrethroids and etofenprox. The selected study sites consisted of two vegetable farming areas, Ekié and Nkolbisson, where insecticide resistance has not yet been assessed, and a third farming area, Nkolondom, serving as a positive control, where insecticide resistance has previously been extensively investigated, including its underlying resistance mechanisms [6,16,19,23,24]. Mosquito species identification revealed the presence of both *An. gambiae* s.s. and *An. coluzzii*, with variable compositions between study sites.

The distribution of *An. gambiae* s.s. and *An. coluzzii* observed in the three study areas is in agreement with previous studies [23,24,25,26], confirming the ubiquity of both species in various ecological settings in Yaoundé. The differences in species composition may be due to temporal shifts in species, owing to biotic interactions occurring at the larval, adult or both stages, such as competition, predation and parasitism [27,28], or by selection pressure resulting from the use of chemical insecticides in both agriculture and public health [23,29].

The susceptibility tests conducted with the 1× DC confirmed pyrethroid resistance in the Nkolondom *An. gambiae* s.s. population, and in the Nkolbisson and Ekié *An. coluzzii* populations. In addition, etofenprox resistance has been observed in both species for the first time and across all three study sites. Etofenprox is a non-ester pyrethroid (pseudo-pyrethroid); this insecticide has been recommended by the WHO as an alternative for indoor residual spraying operations [30] and the treatment of bed nets [3] against malaria vectors because it exhibits lower toxicity to non-target organisms. This active ingredient may be deployed in areas where malaria vectors are susceptible, in order to overcome pyrethroid resistance [31]. However, the development of etofenprox resistance alongside resistance to the ester pyrethroids in *An. gambiae* s.s. and *An. coluzzii* populations from the three study sites emphasizes the need for other alternative vector-control measures in Yaoundé. Moreover, further investigations, including identification of the underlying mechanisms, are required to assess the geographic distribution of etofenprox resistance in the other parts of Cameroon.

The bioassays carried out with 5× DC and 10× DC revealed a mosaic pattern in terms of the intensity of insecticide resistance, in both *An. gambiae* s.s. and *An. coluzzii*, over the 18-month monitoring period, with many fluctuations from moderate to high intensity, and vice versa. In a few cases, the resistance intensity remained as moderate or high over the study period. These patterns might be driven by seasonal changes affecting the expression of insecticide resistance genes in one way or another, or an outcome of stochastic variations between bioassays. Either way, none of the studied populations were neither susceptible to pyrethroids and etofenprox, nor did they exhibit a low resistance intensity, highlighting the extent of the insecticide resistance problem in the Yaoundé urban and peri-urban areas.

Improper use of insecticides in agriculture and public health has been recognized as a major factor leading to the selection of insecticide resistance in malaria vectors, with substantial temporal variations as a function of treatment cycles and seasonality [4,5,6,7,8,9]. Considering the likely selection pressure, which is continuously imposed on the malaria vector populations due to heavy use of insecticides in market gardening areas [32], the observed resistance pattern does not come as a surprise. Indeed, swamps in Yaoundé are subject to urban agriculture permanently operating throughout the year with intensive and improper use of chemical insecticides [24]. Previous studies revealed that agricultural pesticides and mainly insecticide applications are commonly utilized in Nkolondom and Nkolbisson, although the farmers do not receive specific training on pesticide management [32]. Indeed, insecticides of different classes, including pyrethroids (cypermethrin, deltamethrin, and lambdacyhalothrin), carbamates (carbofuran), organophosphates (Dimethoate, Diazinon, and Pyriforce^®^ (chlorpyrifos ethyl)), and organochlorines (Endosulfan), are frequently used by farmers to fight against crop pests in vegetable farms in Yaoundé [32]. In addition, LLINs and other insecticide-based tools are commonly used for malaria prevention and personal protection against mosquito bites in households [33].

A broad analysis of resistance intensity versus resistance frequency showed that a high resistance frequency, as revealed from the substantial decline in mosquito mortality to 1× DC, may not systematically reflect a significant increase in the intensity of resistance obtained from tests with 10× DC. Therefore, the current study clearly demonstrates the added value of resistance intensity assessments in the monitoring of vector resistance to insecticides. We presumed that the mosquito samples falling within the range 0% to 98% mortality rates to 10× DC, indicating high resistance intensity according the WHO criteria [34], may also undergo different levels of high resistance intensity over the time. Therefore, we further classified high resistance into three levels, i.e., level 1 (75–98% mortality rate―HL_1_), level 2 (50–74% mortality rate―HL_2_) and level 3 (<50% mortality rate―HL_3_). This additional classification could be used to follow up the evolution of high resistance intensity in field mosquito populations. Furthermore, based on this supplementary classification, it appears that the HL_3_ resistance intensity recorded with deltamethrin in Nkolbisson and alpha-cypermethrin in Nkolondom is comparable with that reported in *An. gambiae* s.l. from Kolokopé cotton cultivation areas in Togo, with less than 40% mortality at the 10× DC [35]. In the southern and central regions of Mali, high-intensity resistance was reported in 16 *An. gambiae* s.l. populations, fluctuating between HL_1_ and HL_3_ over the test period from 2016 to 2018 [36]. Conversely, in Zambia, Ethiopia, and Nigeria, there was an increase in resistance intensity from moderate to high, or from susceptible to low resistance intensity over the monitoring period [37,38]. However, the variability in the intensity of insecticide resistance observed in *An. gambiae* s.s. and *An. coluzzii* populations from Yaoundé may reflect the situation in several areas of Cameroon, whether resistance has already been reported there or not.

The patterns of resistance intensity reported in this study may result from the simultaneous presence of multiple resistance mechanisms, including target-site and metabolic resistance. We hypothesize that multiple insecticide mechanisms in both *An. gambiae* s.s. and *An. coluzzii* species may have driven the variations of pyrethroid and etofenprox resistance intensity in the surveyed populations, due to the level of selection pressure. Almost half of the analyzed specimens were homozygous-resistant at the *kdr* L995F locus (L995F/L995F), contrary to one L995F/S hybrid in a single *An. gambiae* s.s. individual. In addition to the *kdr* L995F and L995S alleles, the *kdr* N1570Y, also called “super *kdr*”, was found in Nkolondom (frequency = 16.0%). The presence of the N1570Y mutation in Nkolondom is in line with Bamou et al. (recorded frequency = 9.46%) [18]; however, there was a higher allelic frequency in the present study. Combined with the L995F mutation, the N1570Y mutation may reinforce the intensity of *An. gambiae* s.s. resistance to pyrethroids in the study area. The G280S mutation conferring resistance to organophosphates and carbamates was also found in Nkolondom (frequency = 13.0%), confirming previous reports (recorded frequency = 21.78%) [18,39]. Neither N1570Y mutations nor G280S mutations were present in the other two study sites (i.e., Nkolbisson and Ekié). The multiplicity of target-site resistance mutations in *An. gambiae* s.s. from Nkolondom is consistent with previous reports, whereas the presence of the *kdr* L1014F at 60–65% frequencies in Nkolbisson and Ekié is documented here for the first time. In addition, the expression analysis of genes implicated in metabolic resistance, performed in the Nkolondom and Nkolbisson mosquito populations, showed that the P450 genes *Cyp6m2* and *Cyp9k1* were highly upregulated, above fivefold in both populations. Both genes have recently been reported to be upregulated in *An. gambiae* s.l. populations from Cameroon [18]. It has previously been shown that *Cyp9k1* is implicated in the metabolism of the alpha-cyano pyrethroid deltamethrin and *Cyp6m2* in the metabolism of alpha- and non-alpha-pyrethroids [40,41]. *Gste**2*, which confers resistance to DDT [42], was also found to be highly upregulated in Nkolondom and Nkolbisson.

The current study provides valuable data for the evidence-based selection of areas where additional vector-control measures are urgently needed to sustain the ongoing interventions with LLNs.

## 4. Material and Methods

### 4.1. Study Sites

The study was conducted in three vegetable farming areas located in Nkolondom (3°57′18″ N, 11°29′36″ E), Nkolbisson (3°52′46″ N, 11°25′55″ E) and Ekié (3°49′38″ N, 11°31′53″ E), in the north, west and east of Yaoundé, respectively: Yaoundé is the capital city of Cameroon (Figure 1). Nkolbisson is located 8 km from Nkolondom and 13 km from Ekié, whereas the distance between Ekié and Nkolondom is 16 km. Yaoundé lies at 760 m altitude; the average annual temperature is 23.8 °C, with 1628.3 mm average annual rainfall and an annual average 83.0% relative humidity. The climate is characterized by two wet and two dry seasons [43]. The rains mostly fall from March to November, with two peaks in May (219.7 mm) and October (296.1 mm) and a relative breakdown between July and August (short dry season) when precipitation does not exceed 120 mm per month. The main dry season lasts for 3 months, from December to February; the rainfall at that time is less than 50 mm per month. The hydrographic network is very diverse, but consists mostly of streams, rivers, and ponds.

The three study sites are subject to vegetable farming activities due to the presence of swamps (Figure 1). The number of farmers in these districts has increased rapidly since the 2000s, as one of the consequences of the rapid urbanization associated with high demand for food. At least 85% of the population living in these marshlands are involved in agriculture. The food production is diverse, including leafy vegetables, condiments, floriculture, and other food crops highly valued by city dwellers, which are grown year-round [44]. The crop production is intensive and associated with a wide use of chemical fertilizers as well as pesticides [30,44]. More than 70% of farmers treat their crops with at least two chemicals per treatment cycle [28,44]. The puddles created between the cultivated ridges are suitable breeding sites for *Anopheles* mosquitoes, vectors of *Plasmodium* parasites. The mosquito species breeding here belong to the *An. gambiae* species complex and *An. funestus* group. The estimated annual entomological inoculation rates are high, reaching up to 92 infective bites/person/year (ib/m/y) [45]. The transmission is perennial; malaria infections are essentially due to *Plasmodium falciparum*, with few *P. malariae* cases reported [11,46]. The annual *Plasmodium* parasite prevalence in the general populations of these areas ranges from 25% to 55% (unpublished data).

### 4.2. Mosquito Collection, Rearing and Processing

Mosquito collections were conducted in April–May 2018 during the short rainy season, in December 2018–January 2019 during the main dry season, and September–October 2019, during the main rainy season. In each study site, all open water bodies in and around the vegetable farms, including large drain channels, puddles, stream-bed pools, and swamps (Figure 1) were inspected across a total area of ≈ 1–2 km^2^. *Anopheles* larvae and pupae were collected from their breeding sites using the dipping method [47]. Collected mosquito larvae were transported to the Malaria Research Laboratory of Organisation de Coordination pour la lutte contre les Endémies en Afrique Centrale (OCEAC) and reared in the insectary until adult emergence at 27 ± 3 °C, 60–80% relative humidity, and a 12:12 light-dark cycle. Upon emergence, female *An. gambiae* s.l. were identified using morphological identification keys [48,49]. Three- to five-day-old unfed females of *An. gambiae* s.l. were used for insecticide susceptibility and resistance intensity tests, as well as for molecular analysis.

### 4.3. Insecticide Susceptibility Assays

Insecticide resistance was assessed using the standard WHO susceptibility test procedures for adult mosquitoes [34]. Tests were performed under ambient room temperature (24–27 °C) and a relative humidity of 70–80%, using discriminating concentrations (DCs) of insecticides on filter paper. The test DCs and insecticides included three ester pyrethroids, i.e., 0.05% deltamethrin, 0.75% permethrin and 0.05% alpha-cypermethrin, and the non-ester pyrethroid 0.5% etofenprox. Insecticide-impregnated filter papers were supplied by the Vector Control Research Unit (VCRU) of the Universiti Sains of Malaysia. One bioassay included four replicates of 20–25 female mosquitoes from field-collected larvae per insecticide. A total of 3492 mosquitoes were used for these tests, including 1192 from Ekié, 1200 from Nkolondom and 1100 from Nkolbisson (Table 4). One to two tubes of 20–25 mosquitoes were exposed to silicone-oil-impregnated paper and served as controls. After a one-hour exposure, mosquitoes were transferred to holding tubes and provided with a cotton pad soaked in 10% sugar solution for feeding.

The number of mosquitoes knocked down during exposure to insecticide-impregnated papers was recorded at five-minute intervals, and mortality was determined 24 h post-exposure. Tests were also performed with three- to five-day-old unfed females of the insecticide-susceptible *An. gambiae* s.s. Kisumu laboratory colony.

### 4.4. Resistance Intensity Assays

Once the resistance status was determined using the DCs, intensity bioassays with 5× and 10× DCs for alpha-cypermethrin, deltamethrin, permethrin and etofenprox were performed according to the standard WHO bioassay method as described above [34]. All insecticides combined, 3534 and 2556 *An. gambiae* s.l. mosquitoes were used for intensity tests with 5× and 10× DCs, respectively (Table 4).

The filter papers impregnated with 5× DCs and 10× DCs (Table 2) were also supplied by the VCRU. Results were interpreted following the WHO criteria for insecticide resistance intensity [35]. Mosquito specimens were then stored in RNA*later*^®^ (Thermo Fisher Scientific, Waltham, MA, United States) at −20 °C for further molecular analysis.

### 4.5. Molecular Assays

A subset of 50 *An. gambiae* s.l. specimens per study site, used as controls during the susceptibility and resistance intensity bioassays conducted in April 2018, were randomly selected and used for species identification, *kdr* L995F, L995S, and N1570Y, and *Ace-1* G280S genotyping. These analyses were carried out at the molecular biology laboratory of the Centre Suisse de Recherches Scientifiques en Côte d’Ivoire (CSRS) in Abidjan, Côte d’Ivoire. Genomic DNA was extracted from individual mosquitoes using a magnetic beads-based protocol (MagnaMedics Diagnostics GmbH, Aachem, Germany, Cat. No. MD01017), following [50], and stored at −20 °C until used for the TaqMan^®^ reverse-transcription quantitative polymerase chain reaction (RT-qPCR) assays. The lysate was diluted 10× in PCR-grade water and used for the RT-qPCR SNP TaqMan^®^ assays (Eurofins Genomics, Ebersberg, Germany). Species identification and detection of insecticide resistance mutations were performed on the CFX96 Bio-Rad Real-Time PCR system (Bio-Rad Laboratories Inc, Hercules, CA, USA) using a one-step RT-PCR master mix supplied by FTD (Fast-track diagnostics, Luxembourg) in a total reaction volume of 10 µL [50,51,52,53,54].

Another subset of 50 randomly selected *An. gambiae* s.l. specimens per study site also collected in April–May 2018 was used for metabolic and cuticular resistance gene expression analysis at the OCEAC laboratory of medical entomology. Expression levels of *Cyp6p3*, *Cyp6m2*, *Cyp9k1*, *Cyp6p4*, *Cyp6z1*, *Cyp6p1*, and *Cyp4g16* were measured relative to a housekeeping gene encoding the ribosomal protein S7 (RPS7) in field-collected specimens and compared to expression levels of the same detox genes in the susceptible Kisumu colony using triplex assays with the direct RT-qPCR approach: 1:200 diluted lysates without RNA extraction [55]. Briefly, 10 mosquitoes were homogenized in 100 µL of RTL lysis buffer (QIAGEN, Cat. No. 79216) using a battery-powered tissue grinder and a plastic pestle in a 1.5 mL micro centrifuge tube. The lysate was diluted using RNase-free water (1:200), and 10 µL was directly added to “ready-to-go” lyophilized pellets presented in Vontas and Mavridis, 2019 [56]. The runs were performed in a QuantStudio™ 5 Real-Time PCR System (Applied Biosystems™, Waltham, MA, USA) with the following thermal cycle parameters: 50 °C for 15 min, 95 °C for 3 min, and 40 cycles of 95 °C for 3 s and 60 °C for 30 s. Samples were amplified in at least two technical replicates, using three biological replicates for gene expression analysis for each population. QuantStudioTM Design & Analysis Software v1.5.2 (Applied Biosystems™, Waltham, MA, USA) was used for the calculation of Ct values for each reaction, which were then used to calculate fold-changes according to the Pfaffl method [57].

Assays were designed and performed in the framework of the interdisciplinary research project DMC-MALVEC (https://dmc-malvec.eu, accessed on 8 September 2021). The list of probes and primers used for molecular analysis is provided in Table 5.

### 4.6. Data Analysis

The resistance status was classified according to the WHO criteria [34]. Mortality rates below 90% indicate resistance, whereas those above 98% indicate susceptibility. Mortality rates between 90% and 98% indicate possible resistance to be confirmed. For assessment of the resistance intensity, the following WHO interpretation criteria [34] were used:

Mortality rates of 98–100% at 5× DC indicate a low resistance intensity;

Mortality rates less than 98% at 5× DC indicate a moderate to high resistance intensity, and 98–100% mortality rates at 10× DC confirm a moderate resistance intensity;

Mortality rates of less than 98% at 10× DC indicate a high resistance intensity.

The trends of mortality rates as an indicator of resistance status or resistance intensity were analyzed using the *R* statistical software (R Core Team 2021, version 4.1.1) [58] via *RStudio*. Packages *dplyr, ggplot2* were used for data description and generating graphs of mortality rates of tested *An. gambiae* s.l. samples [59], whereas *lme4* and *lmerTest* were used for modelling the resulting profile of insecticide resistance [60,61].

We considered logistic regression with the logit link function to model mortality as an outcome. In the experiments performed, the mosquitoes were grouped in tubes. Therefore, we first confronted the baseline classical logistic model with the mixed-effects model in which the randomized variable was the tube. For this, we performed the log-likelihood ratio test. The test was significant at the 5% level, with a *p*-value ≈ 0 < 1% (statistic ≈3750, degrees of freedom = 1). Therefore, the mortality data were modeled using mixed-effects logistic regression. Thus, the remaining mortality analyses were based on the mixed-effects logistic regression model in which the randomized variable was the tube. The fixed effects were the insecticide and mosquito collection period.

Statistical analyses were carried out using the statistical software *R* via *RStudio*. In particular, the following packages were used: *dplyr*, *ggplot2* for data description and generating graphs, whereas *lme4* and *lmerTest* were used for modelling mortality.

## 5. Conclusions

This study provides evidence of moderate to high levels of resistance intensity to ester pyrethroids and the non-ester pyrethroid etofenprox in vegetable farming areas in Yaoundé, Cameroon. Therefore, the effectiveness of LLIN treated with ester and non-ester pyrethroids may be compromised in these areas. Additionally, IRS with etofenprox is not a good alternative for the replacement of LLINs in these areas. Further investigations are needed to assess the geographic distribution of the high resistance intensity, and the underlying mechanisms in other areas in Cameroon.

## Figures and Tables

**Figure 1 molecules-26-05543-f001:**
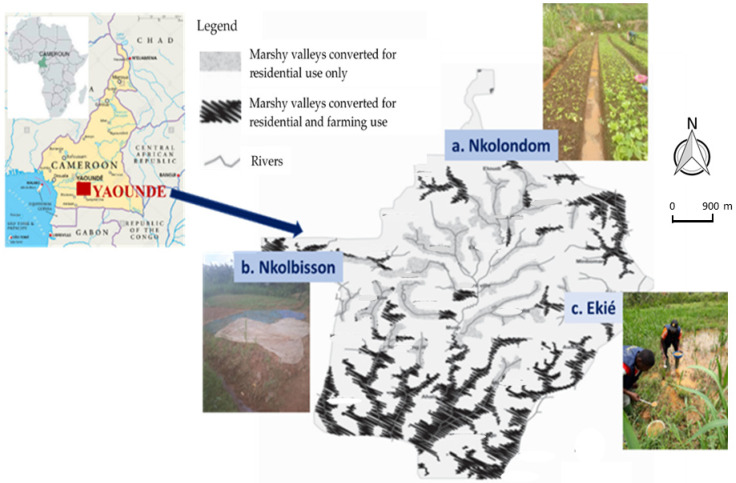
Map of Yaoundé in Cameroon showing the marshy valleys with typical mosquito breeding sites in Nkolondom, Nkolbisson and Ekié, where immature *Anopheles gambiae* s.l. stages were collected (a large drain channel in Nkolondom; stream-bed pool in Nkolbisson; swamp in Ekié). (Adapted from [22]).

**Figure 2 molecules-26-05543-f002:**
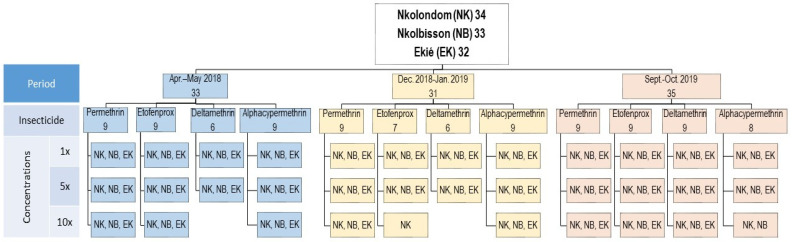
Study design showing resistance incidence and resistance intensity assays carried out with field-collected *Anoheles gambiae* s.l. samples throughout the two-year study period (2018–2019). (Apr.–May 2018: April–May 2018; Dec. 2018–Jan. 2019: December 2018–January 2019; Sept.–Oct. 2019: September–October 2019).

**Figure 3 molecules-26-05543-f003:**
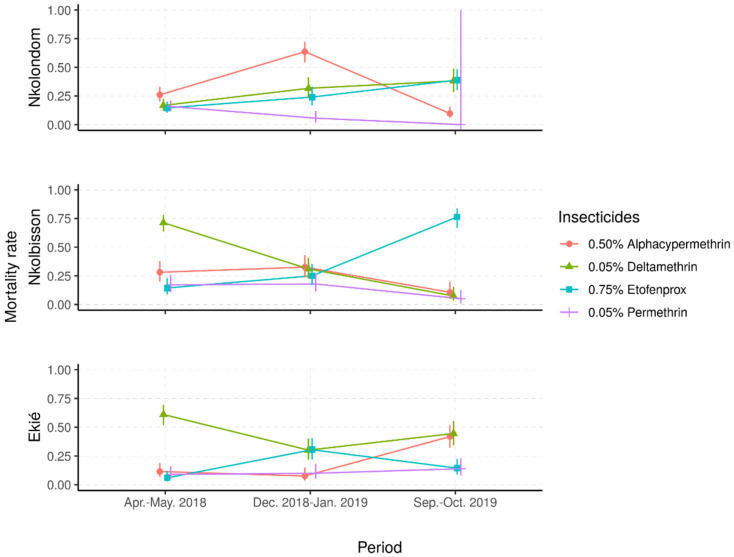
Predicted mortality rates and their 95% confidence intervals from a mixed-effects logistic regression model of field *Anopheles gambiae* s.l. samples following exposure to four insecticide diagnostic concentrations. (Apr.–May 2018: April–May 2018; Dec. 2018–Jan. 2019: December 2018–January 2019; Sept.–Oct. 2019: September–October 2019).

**Figure 4 molecules-26-05543-f004:**
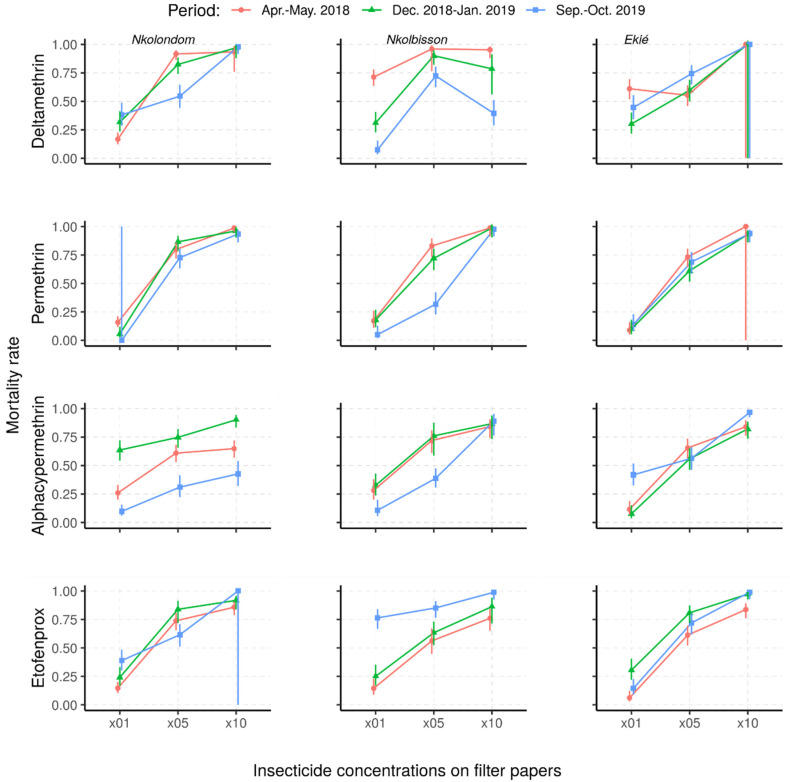
Predicted mortality rates and their 95% confidence intervals of field-collected *Anopheles gambiae* s.l. samples from a mixed-effects logistic regression model following exposure to 1×, 5× and 10× insecticide concentrations (Apr.–May 2018: April–May 2018; Dec. 2018–Jan. 2019: December 2018–January 2019; Sept.–Oct. 2019: September–October 2019).

**Table 1 molecules-26-05543-t001:** Frequencies of *kdr* L995F/S and N1570Y and *Ace-1* G280S resistance alleles among *Anopheles gambiae* s.s. and *An. coluzzii* populations from Nkolondom, Nkolbisson and Ekié, Cameroon, in April 2018.

Colony	n	Species	(%)				Genotype												
				*kdr* L995F/S						*kdr* N1570Y					*Ace-1* G280S				
				n	SS	RwS	RwRw	RwRe	Freq	n	SS	SR	RR	Freq	n	SS	SR	RR	Freq
Nkolondom	50	*Anopheles gambiae* s.s.	100	50	1	26	22	1	0.72	32	22	10	0	0.16	50	37	13	0	0.13
Nkolbisson	50	*Anopheles gambiae* s.s.	4	2	0	2	0	0	0.50	−	−	−	−	−	2	2	0	0	0
		*Anopheles coluzzii*	96	48	0	31	17	0	0.65	30	30	0	0	0	48	48	0	0	0
Ekié	50	*Anopheles coluzzii*	100	50	0	40	10	0	0.6	30	30	0	0	0	50	50	0	0	0
Kisumu	50	*Anopheles gambiae* s.s.	100	50	50	0	0	0	0.0	30	30	0	0	0	50	50	0	0	0

n: number of specimens tested; SS: homozygote-susceptible; RwS: heterozygote LF; RwRw: homozygote *kdr* FF; RwRe: heterozygote *kdr* FS; Freq: allelic frequency; SR: heterozygote; RR: homozygote-resistant.

**Table 2 molecules-26-05543-t002:** Patterns of resistance frequency versus resistance intensity among *Anopheles gambiae* s.l. populations from Nkolondom, Nkolbisson and Ekié during the three surveys.

Insecticide	Colony	Insecticide Concentration (%)	Resistance Variable	Mortality Rate (%) per Survey Period		
				Apr.–May 2018	Dec. 2018–Jan. 2019	Sept.–Oct. 2019
Deltamethrin	Kisumu	0.05_dc_	F	100	99	100
	Nkolondom	0.05_dc_	F	16.0	31.7	45.2
		0.5_10×_	I			
	Nkolbisson	0.05_dc_	F	56.7	31.0	4.9
		0.5_10×_	I			
	Ekié	0.05_dc_	F	61.1	30.1	44.6
		0.5_10×_	I			
Permethrin	Kisumu	0.75_dc_	F	98.7	100	99
	Nkolondom	0.75_dc_	F	8.7	5.6	0.0
		7.5_10×_	I			
	Nkolbisson	0.75_dc_	F	16.3	17.9	5.1
		7.5_10×_	I			
	Ekié	0.75_dc_	F	7.2	10.0	13.9
		7.5_10×_	I			
Alpha-cypermethrin	Kisumu	0.05_dc_	F	100	100	100
	Nkolondom	0.05_dc_	F	17.8	63.7	3.2
		0.5_10×_	I			
	Nkolbisson	0.05_dc_	F	17.8	32.6	10.7
		0.5_10×_	I			
	Ekié	0.05_dc_	F	11.5	7.5	41.8
		0.5_10×_	I			
Etofenprox	Kisumu	0.5_dc_	F	100	100	99
	Nkolondom	0.5_dc_	F	9.4	24.1	38.9
		5.0_10×_	I			
	Nkolbisson	0.5_dc_	F	14.3	25.0	76.3
		5.0_10×_	I			
	Ekié	0.5_dc_	F	6.0	30.4	14.4
		5.0_10×_	I			
**Resistance intensity**		**Moderate**	**High Level I**	**High Level II**	**High Level III**	
Mortality rate to 10×		>98%	76–98%	50–75%	<50%	

F: resistance frequency (%); dc: standard insecticide diagnostic concentration; I: resistance intensity; 10×: tenfold insecticide diagnostic concentration; Kis. strain: Kisumu strain of *Anopheles gambiae* s.s.; Apr.–May 2018: April–May 2018; Dec. 2018–Jan. 2019: December 2018–January 2019; Sept.–Oct. 2019: September–October 2019.

**Table 3 molecules-26-05543-t003:** Differential expression of metabolic and cuticular resistance genes among the Nkolondom and Nkolbisson mosquito populations as compared with the Kisumu susceptible strain in April 2018.

Gene	Mechanism	Nkolondom	Nkolbisson
*Cyp6p3*	Metabolic Resistance	0.4 (0.2–1.1)	1.5 (0.8–3.8)
*Cyp6m2*		17.1 (5.9–47.5) *	18.9 (5.7–56.6) *
*Cyp9k1*		6.0 (2.94–13.5) *	6.8 (3.6–15.3) *
*Cyp6p4*		2.9 (1.8–4.4) *	6.2 (4.7–9.3) *
*Cyp6z1*		2.6 (1.8–3.6) *	3.4 (2.3–4.5) *
*Gste2*		12.8 (7.9–23.2) *	52.0 (29.2–93.8) *
*Cyp6p1*		0.8 (0.4–1.6)	2.1 (0.9–5.1)
*Cyp4g16*	Cuticular Resistance	2.4 (2.1–2.7) *	2.5 (1.9–3.2) *

Asterisks (*) indicate statistically significant. Differential expression values refer to fold changes in gene expression (95% CI).

**Table 4 molecules-26-05543-t004:** Number of mosquitoes used for insecticide susceptibility and resistance intensity testing.

Insecticides	Collection Period	1× DC			5× DC			10× DC			N
		Ekié	Nkolondom	Nkolbisson	Ekié	Nkolondom	Nkolbisson	Ekié	Nkolondom	Nkolbisson	
Delta	Apr.–May 2018	113	106	106	124	91	84	MD	MD	MD	624
	Dec. 2018–Jan. 2019	93	108	100	102	101	91	MD	MD	MD	595
	Sept.–Oct. 2019	83	90	81	80	84	142	101	88	81	830
Perm	Apr.–May 2018	111	103	104	112	81	85	118	83	114	911
	Dec. 2018–Jan. 2019	90	107	95	103	104	86	101	104	97	887
	Sept.–Oct. 2019	86	99	87	90	92	89	MD	91	88	722
Alpha	Apr.–May 2018	113	101	83	115	106	116	110	84	89	917
	Dec. 2018–Jan. 2019	93	110	89	102	103	99	101	85	97	879
	Sept.–Oct. 2019	98	85	80	94	87	85	MD	94	94	717
Etofenprox	Apr.–May 2018	116	96	98	116	83	123	122	80	87	921
	Dec. 2018–Jan. 2019	92	104	84	99	93	89	MD	89	90	740
	Sept.–Oct. 2019	104	91	93	81	108	94	93	80	95	839
N		1192	1200	1100	1218	1133	1183	746	878	932	9582

MD: Missing data; Delta: Deltamethrin; Perm: Permethrin; Alpha: Alpha-cypermethrin; N: Total number of mosquitoes. (Apr.–May 2018: April–May 2018; Dec. 2018–Jan. 2019: December 2018–January 2019; Sept.–Oct. 2019: September–October 2019).

**Table 5 molecules-26-05543-t005:** List of probes and primers used for molecular analysis.

Assayed Marker	Oligonucleotides Name	Sequence	Assay Name
Species Identification	S200-6.1 F	TCGCCTTAGACCTTGCGTTA	Molecular Forms
	S200-6.1 R	CGCTTCAAGAATTCGAGATAC	Molecular Forms
	AgM-P	ACCGCGCCGCCATACGTAGGA	*An. coluzzii*
	AgS-P	ATGTCTAATAGTCTCAATAGT	*An. gambiae*
Kdr L995 mutation	Kdr-F	CATTTTTCTTGGCCACTGTAGTGAT	kdr
	Kdr-R	CGATCTTGGTCCATGTTAATTTGCA	kdr
	kdrWT-P	CTTACGACTAAATTTC	Wild type-kdr
	kdrRw-P	ACGACAAAATTTC	West-kdr
	kdrRe-P	ACGACTGAATTTC	East-kdr
Kdr N1570Y mutation	1575-F	TGGATCGCTAGAAATGTTCATGACA	Kdr+
	1575-R	CGAGGAATTGCCTTTAGAGGTTTCT	Kdr+
	N1575-P	ATTTTTTTCATTGCATTATAGTAC	Wild type-kdr+
	Y1575-P	TTTTTCATTGCATAATAGTAC	Mutant-kdr+
Ace1 G280S mutation	ACE1-F	GGCCGTCATGCTGTGGAT	iAChe
	ACE1-R	GCGGTGCCGGAGTAGA	iAChe
	Ace 1G_WT-P	TTCGGCGGCGGCT	Wild type-iAChe
	Ace 1G_MT-P	TTCGGCGGCAGCT	Mutant-iAChe
Normalizer	*Rps7*F	CCACCATCGAACACAAAGTTGA	(A)-(D) [RG]
	*Rps7*-R	TGCTGCAAACTTCGGCTATTC	(A)-(D) [RG]
	*Rps7*-P	FAM-CCGTGACGTTACGTTCGAATTCCCA-BHQ1	(A)-(D) [RG]
Detoxification Enzyme(metabolic resistance)	*Cyp6p3*-F	ACAATGTGATTGACGAAACCCT	(A)
	*Cyp6p3*-R	GGATCACATGCTTTGTGCCG	(A)
	*Cyp6p3*-P	HEX-ACCCGCGTACCGTCTGTGGACT-BHQ1	(A)
	*Cyp6m2*-F	CTGGCGTTGAATCCAGAGGT	(A)
	*Cyp6m2*-R	GATACTTGCGCAGTGATTCATTAAG	(A)
	*Cyp6m2*-P	ATTO647N-AGAGAAATCCTGCAAAAGCACAACGGAGA-BHQ3	(A)
	*Cyp9k1*-F	CCGACACGTGGTGATGGATAC	(B)
	*Cyp9k1*-R	CGTCGTCGGTCCAGTCAAC	(B)
	*Cyp9k1*-P	HEX-CAATCTTCTGATGCAGGCCCGCAA-BHQ1	(B)
	*Cyp6p4*-F	CTGGACAACGTTATCAATGAAACC	(B)
	*Cyp6p4*-R	GCACGGTGTAATCACGCATC	(B)
	*Cyp6p4*-P	ATTO647N-CCGATCGAGTCACTTTCGCGCG-BHQ3	(B)
	*Cyp6z1*-F	CCCGCAACTGTATCGGTCTG	(C)
	*Cyp6z1*-R	TTCGGTGCCAGTGTGATTGA	(C)
	*Cyp6z1*-P	HEX-TGATGCTGTCCCGATTTAACTTTTCGGC-BHQ1	(C)
	*Gste2*-F	CCGGAATTTGTGAAGCTAAACC	(C)
	*Gste2*-R	GCTTGACGGGGTCTTTCGG	(C)
	*Gste2*-P	ATTO647N-CGGTACGATCATCACCGAGAGCCAC-BHQ3	(C)
	*Cyp6p1*-F	ACAGGTGGTGAACGAAACCC	(D)
	*Cyp6p1*-R	GGTGTAATCCTGTCCCGCAA	(D)
	*Cyp6p1*-P	HEX-CCGCTCGAAACGACGCTGCG-BHQ1	(D)
Cuticular hydrocarbon synthesis (cuticular resistance)	*Cyp4g16*-F	GTCCAAGAAGTTGCGTCGGAC	(D)
	*Cyp4g16*-R	TCTTCGATTTGCGTTGACGTG	(D)
	*Cyp4g16*-P	ATTO647N-CTGCAGGCCGACATCATTTTGAAGC-BHQ3	(D)

F: Forward primer; R: Reverse primer; P: TaqMan probe; RG: Reference gene.

## Data Availability

All data generated or analyzed during the current study are included in this published article.

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
