# Peer review of "Pyrethroid and Etofenprox Resistance in Anopheles gambiae and Anopheles coluzzii from Vegetable Farms in Yaoundé, Cameroon: Dynamics, Intensity and Molecular Basis"

_molecules, 2021, doi:10.3390/molecules26185543_

Round 1
Reviewer 1 Report
Review of the MS - Molecules “Pyrethroid and etofenprox resistance in Anopheles gambiae and An. coluzzii from vegetable farms in Yaoundé, Cameroon: dynamics, intensity and molecular basis"
The authors report specific data on anthropophilic Anopheles species in a tropical wet area of Cameroon and unfortunately, Yaoundé features a lengthy wet season, constant temperatures that favors Anopheles species as An. gambiae and An. coluzzii. The MS presented with the subject matter is interesting and can help provide knowledge to scholars in the sector and to the governing bodies for mitigation interventions for related socio-health issues. The extensive use of pyrethroids also in horticultural crops elicit different levels of resistance to numerous populations beyond the geographic isolation of these populations. Despite the emergence and spread of mosquito resistance to pyrethroids, insecticide-treated nets continue to provide a substantial level of protection in most settings.
Although the MS is interesting in the overall approach to the problem, it requires some simplification interventions on the text that in some passages is complex in the succession of activities carried out and sometimes difficult to follow. A further improvement of the MS could be carried out through a more clear statistical analysis of the temporal findings and more detailed analysis of data could be interesting.
Specific comment
Section 2 Results 2.1. “Species composition” Specify whether the 50 adult mosquito specimens from each study site were collected only in one period or during the three surveys.
Section 2 Results 2.1. 2.2. “Trends in insecticide resistance in Anopheles gambiae s.l. populations.” In my opinion, this section could be moved to Materials and Methods as it reports the number of samples of each site and the test scheme adopted by the authors.
Section 4 Materials and Methods. Specify better if the presence of the species (An. funestus) was detected in the three sampling sites. Is it on all three sites at the same time?.
Section 4 Materials and Methods. Enter the distance between the three sites indicated. From my first visualization on the map the distances are not high, I think less than 20 km.
Section 4 Materials and Methods -sub 4.3. “Insecticide susceptibility assays” “were exposed to each insecticide and concentration.” As the concentration for each active ingredient was previously reported, remove "concentration".
Section 4 Materials and Methods. The specimens Kisumu An. gambiae s.s. colony used were of the same age and status (three- to five-day-old unfed females).
Section 4 Materials and Methods. sub 4.3 “Resistance intensity assays”. Indicate here the number of samples used for the different concentrations of the pyrethroid molecules used.
Section 4 Materials and Methods. sub 4.5. Specify if An. gambiae specimens used for gene expression analysis at the OCEAC lab was collected too in April. This because larval rearing conditions could change the insecticide susceptibility of Anopheles mosquitoes.
Section 4 Materials and Methods. sub 4.6. In the paragraph on statistical analyses such as mixed models, indicate more clearly the randomized variable (eg sampling sites ??). The paragraph above all in the initial part must be reformulated,
Section 4 Materials and Methods. sub 4.6. Report better the choice made with likelihood ratio test. Example. The data is x.x, 95% CI [x.x, x.x] times more likely under mixed model than under logistic regression model. The hypothesis that the data is equally likely under the two models was rejected with p = 0.00x.
Author Response
We are grateful to the reviewers for their valuable comments which help us to improve the quality of the manuscript. Most of the issues raised have been addressed; for those that we have not fully considered, we provided some explanations.
Reviewer 1
The authors report specific data on anthropophilic Anopheles species in a tropical wet area of Cameroon and unfortunately, Yaoundé features a lengthy wet season, constant temperatures that favours Anopheles species as An. gambiae and An. coluzzii. The MS presented with the subject matter is interesting and can help provide knowledge to scholars in the sector and to the governing bodies for mitigation interventions for related socio-health issues. The extensive use of pyrethroids also in horticultural crops elicit different levels of resistance to numerous populations beyond the geographic isolation of these populations. Despite the emergence and spread of mosquito resistance to pyrethroids, insecticide-treated nets continue to provide a substantial level of protection in most settings.
Although the MS is interesting in the overall approach to the problem, it requires some simplification interventions on the text that in some passages is complex in the succession of activities carried out and sometimes difficult to follow.
Comment: A further improvement of the MS could be carried out through a clearer statistical analysis of the temporal findings and more detailed analysis of data could be interesting.
ANSWER: Data were mostly analysed according to the WHO criteria. To classify the vector populations as Resistant or Susceptible, we first referred to the WHO criteria. Then we applied the Likelihood ratio test to study the temporal evolution of insecticide resistance in each site. The detailed description of the analysis methods is given in the text. Other methods could probably be used, but we think the results obtained from the likelihood ratio test also provided good results. We were able to assess the temporal trends and draw conclusions that seem relevant.
Specific comment
Section 2 Results 2.1. “Species composition” Specify whether the 50 adult mosquito specimens from each study site were collected only in one period or during the three surveys.
ANSWER: The 50 adult mosquito specimens from each study site were collected only in the period April-May 2018. The period has been added in the text.
Section 2 Results 2.1. 2.2. “Trends in insecticide resistance in Anopheles gambiae s.l. populations.” In my opinion, this section could be moved to Materials and Methods as it reports the number of samples of each site and the test scheme adopted by the authors.
ANSWER: We agree with this comment. However, according to the instructions to the authors regarding the sections of the manuscript, we had to present the “Results” before the “Methodology”. In order to facilitate the understanding of the results we found helpful to briefly describe the composition of our samples and the insecticide tests performed in the “Results” section before presenting the resistance data.
Section 4 Materials and Methods. Specify better if the presence of the species (An. funestus) was detected in the three sampling sites. Is it on all three sites at the same time?
ANSWER: The anophelines species in the city of Yaoundé and the surrounding areas are mainly composed of 2 species of Anopheles gambiae complex (An. gambiae and An. coluzzii) and species of the Anopheles funestus group (Antonio Nkondjio et al., 2019, Cohuet et al., 2004; Djamouko-Djonkam et al., 2020). To make sure the specimens used for susceptibility tests belonged to the An. gambiae complex, all tested mosquitoes were morphologically identified prior to exposure to insecticides. Indeed, larvae collected from the field were reared to the adult stage and the identification of the emerging adults showed that our samples were composed exclusively of specimens of Anopheles gambiae s.l.; there was no An. funestus. The morphological identification of the mosquitoes tested was confirmed by molecular analysis.
Section 4 Materials and Methods. Enter the distance between the three sites indicated. (From my first visualization on the map the distances are not high, I think less than 20 km.
ANSWER: Nkolbisson is located at 8 km from Nkolondom and 13 km from Ekie, while the distance between Ekie and Nkolondom is 16 km. This information has been added in the manuscript.
Section 4 Materials and Methods -sub 4.3. “Insecticide susceptibility assays” “were exposed to each insecticide and concentration.” As the concentration for each active ingredient was previously reported, remove "concentration".
ANSWER: The text has been revised accordingly.
Section 4 Materials and Methods. The specimens Kisumu An. gambiae s.s. colony used were of the same age and status (three- to five-day-old unfed females).?
ANSWER: Yes, the Anopheles gambiae Kisumu strain which is susceptible to all insecticides is maintained at our Laboratory since more than 25 years.
During each cross-sectional resistance survey, eggs were incubated in trays containing natural water, then pupae were collected each day and kept in cages labelled with the date of adult emergence. Labelled cages with mosquitoes 3 to 5 days old were selected for susceptibility testing to ensure that all mosquitoes used for the bioassays were of the same age range, either from field collections or from laboratory colony.
The age of the Kisumu strain mosquitoes used has been added in the text.
Section 4 Materials and Methods. sub 4.3 “Resistance intensity assays”. Indicate here the number of samples used for the different concentrations of the pyrethroid molecules used.
ANSWER: For each bioassay, 80 to 100 female mosquitoes were used according to the WHO protocols (WHO, 1998; WHO, 2013; WHO, 2016) Detailed sample sizes used has been added in Table 4.
Section 4 Materials and Methods. sub 4.5. Specify if An. gambiae specimens used for gene expression analysis at the OCEAC lab was collected too in April. This because larval rearing conditions could change the insecticide susceptibility of Anopheles mosquitoes.
ANSWER: The Anopheles gambiae specimens used for gene expression analysis at the OCEAC Lab were collected at the same period (April 2018). The platform for characterization of enzymatic markers was not available in Côte d'Ivoire at the time when target site mutations were screened. That is why the diagnosis of enzymes was performed in Cameroon and that of mutations in Côte d'Ivoire. We have revised this part of the manuscript according to your suggestions.
Section 4 Materials and Methods. sub 4.6. In the paragraph on statistical analyses such as mixed models, indicate more clearly the randomized variable (eg sampling sites ??). The paragraph above all in the initial part must be reformulated,
ANSWER: The paragraph on statistical analysis has been revised to indicate the randomized variables, i.e., tubes, insecticides and mosquito collection period.
Section 4 Materials and Methods. sub 4.6. Report better the choice made with likelihood ratio test. Example. The data is x.x, 95% CI [x.x, x.x] times more likely under mixed model than under logistic regression model. The hypothesis that the data is equally likely under the two models was rejected with p = 0.00x.
ANSWER: The criteria for choosing the likelihood ratio test have been clarified in the Data analysis Section.
Reviewer 2 Report
After careful revision of the manuscript it was found that the manuscript covers an important topic and worthy of study and can be published after minor revision which does not affect the quality of the manuscript. Please, considering the specific comments, I would like to suggest:
1 - Some statistics treatment could be reported for data in Figure 4.
I believe that some references could enrich this manuscript, for instance:
- Molecular Simulation , Vol. 43, 121-133 (2017).
- Animals 11(7), 1880 (2021)
- Journal of Exposure Science & Environmental Epidemiology Vol. 31, 549-559 (2021)
Author Response
We are grateful to the reviewers for their valuable comments which help us to improve the quality of the manuscript. Most of the issues raised have been addressed; for those that we have not fully considered, we provided some explanations.
Reviewer 2
After careful revision of the manuscript it was found that the manuscript covers an important topic and worthy of study and can be published after minor revision which does not affect the quality of the manuscript. Please, considering the specific comments, I would like to suggest:
1 - Some statistics treatment could be reported for data in Figure 4.
I believe that some references could enrich this manuscript, for instance:
- Molecular Simulation , Vol. 43, 121-133 (2017).
- Animals 11(7), 1880 (2021)
- Journal of Exposure Science & Environmental Epidemiology Vol. 31, 549-559 (2021)
ANSWER: Thanks for the suggestions.
Figure 4 has been revised to include statistical data as suggested. Regarding references; although they refer to the insecticides used in our study, we unfortunately did not find a close link between the potential impact of insecticides on non-target organisms and the problem of insecticide resistance in malaria vectors that is the subject of the current paper. Nevertheless, we found these articles very interesting and helpful for our further research.
Reviewer 3 Report
Here, Piameu et al. analyzed the intensity of insecticide resistance of Anopheles populations in three farms of to Cameroon in terms of resistance incidence and intensity assessment, detection of mutation frequency and expression profiles of metabolic resistance genes. The study was well designed and conducted, however, I have several minor concerns and will support it publish if the manuscript is improved.
Abstract: With the data in hand, what's the exact mechanisms of high resistance intensity for the three places? A clear conclusion should be given in this part, so readers can quick grasp the the point of the study.
Background: Since the authors have conducted the experiments in three time points, what's the weather like in Apr-May 2018 and Sept-Oct 2019? Climate change might induce the dynamic population composition, then affect the resistance intensity? At least, details of the climate are needed in this part.
Methods: For RT-qPCR, the primer sequences and reference genes are not given, so more information should be clarified using tables or figures.
Results: I think the figure 1 should be given and cited at first in the whole body of the manuscript. Figure3 and 4 are not well illustrated with high resolution pictures and in the figure legend, no information of error bar and statistical significance were supplied.
There are no line numbers in each pages, so it's quite inconvenient for review.
Author Response
We are grateful to the reviewers for their valuable comments which help us to improve the quality of the manuscript. Most of the issues raised have been addressed; for those that we have not fully considered, we provided some explanations.
Reviewer 3
Here, Piameu et al. analyzed the intensity of insecticide resistance of Anopheles populations in three farms of to Cameroon in terms of resistance incidence and intensity assessment, detection of mutation frequency and expression profiles of metabolic resistance genes. The study was well designed and conducted, however, I have several minor concerns and will support it publish if the manuscript is improved.
Abstract: With the data in hand, what's the exact mechanisms of high resistance intensity for the three places? A clear conclusion should be given in this part, so readers can quick grasp the the point of the study.
ANSWER: Several mechanisms were detected in all the three sites, particularly kdr 995F, and Cyp6m2 and Gste2 (which are more than 12 times over-expressed). Therefore we think multiple resistance mechanisms are involved (target mutation and increased detoxification enzyme) as mentioned in the abstract.
Background: Since the authors have conducted the experiments in three time points, what's the weather like in Apr-May 2018 and Sept-Oct 2019? Climate change might induce the dynamic population composition, then affect the resistance intensity? At least, details of the climate are needed in this part.
ANSWER: The study was conducted during the small rainy season (April-May 2018), the main dry season (December 2018 and January 2019) and the main rainy season (September-October 2019). This information has been added in the “Background” Section (Page 3) Thanks for this useful comment.
Methods: For RT-qPCR, the primer sequences and reference genes are not given, so more information should be clarified using tables or figures.
ANSWER: Table 5 showing the primer sequences has been added to the manuscript.
Results: I think the figure 1 should be given and cited at first in the whole body of the manuscript. Figure3 and 4 are not well illustrated with high resolution pictures and in the figure legend, no information of error bar and statistical significance were supplied.
ANSWER: Figure 1 has been placed before the “Results” Section and reference provided all over the text.
The information on error bars and statistical significance has been added in data analysis sub-section. We also revised the figures 3 and 4 to improve their resolution.
There are no line numbers in each pages, so it's quite inconvenient for review.
ANSWER: The line numbers were included in the submitted manuscript. These lines were probable removed when editing the text before they sent it to you for review. Sorry for the inconvenience.
Round 2
Reviewer 1 Report
You have accepted the notes and indications reported in the previous version of the MS. I reiterate that the MS is very interesting and I encourage you to continue improving your research activities in this area.